# Statistical and Microstructural Analyses of Al–C–Cu Composites Synthesized Using the State Solid Route

**DOI:** 10.3390/ma14081969

**Published:** 2021-04-14

**Authors:** Audel Santos Beltrán, Verónica Gallegos Orozco, Miriam Santos Beltrán, Cynthia Gómez Esparza, Iza Ronquillo Ornelas, Carmen Gallegos Orozco, Luz. E. Ledezma Beng, Roberto Martínez Sánchez

**Affiliations:** 1Departamento de Nanotecnología, Universidad Tecnológica de Chihuahua Sur, Km. 3.5 Carr., Chihuahua a Aldama, Chihuahua 31313, Mexico; vgallegos@utchsur.edu.mx (V.G.O.); msantos@utchsur.edu.mx (M.S.B.); ironquillo@utchsur.edu.mx (I.R.O.); 2Centro de Investigación en Materiales Avanzados (CIMAV), Departamento de Metalurgia e Integridad Estructural, Miguel de Cervantes No. 120, Chihuahua 31109, Mexico; cynthiadeisy@gmail.com (C.G.E.); roberto.martinez@cimav.edu.mx (R.M.S.); 3Departamento de Ciencias Básicas, Tecnológico Nacional de México, Campus Chihuahua II, Ave. de las Industrias #11101, Complejo Industrial Chihuahua, Chihuahua 31130, Mexico; carmen.go@chihuahua2.tecnm.mx (C.G.O.); lucybeng12@gmail.com (L.E.L.B.)

**Keywords:** Al-based composites, mechanical milling, nanoparticles dispersion

## Abstract

Aluminum powder with different C and C–Cu mixtures powders were fabricated by powder metallurgy, using high-energy mechanical milling as a pre-treatment of powders. To evaluate the combined effect of the C–Cu mixture and the process conditions, such as sintering temperature/time and milling time, on the yield stress and hardness, two experimental designs were carried out. The results were analyzed with Minitab Statistical Software using contour plots. From the results, better mechanical properties were found at a Cu/C ratio of 0.33 and samples with high C content (3 wt. %). In samples subject to long sintering time (3 h), the mechanism of precipitation of the second phase was mainly present, resulting in an improvement in mechanical properties. From the difference found between the elastic limit and the microhardness tests, it was found that there was an inefficient sintering process affecting the elastic limit test results. Additionally, X-ray analyses using the Rietveld program, were used for microstructural characterization and mechanical parameters of yield strength and ultimate tensile strength.

## 1. Introduction

Metal matrix composites (MMC) are characterized by having different properties that include high thermal resistance, low coefficient of thermal expansion, high specific resistance, sound damping capabilities, high resistance to corrosion, and high specific stiffness [1,2]. Ceramic-reinforced metal matrix composites are designed to combine the different mechanical properties of different materials to obtain a composite material for specific industrial applications. Some of the most used particles [3] that are used in the manufacture of MMC as reinforcements are silicon carbide (SiC), aluminum oxide (Al_2_O_3_), zirconia (ZrO_2_), boron carbide (B_4_C), and graphite (Gram). Ajay R. Bhardwaj et al. [4,5] discussed various machinability aspects of MMC. Aluminum-based compounds have been manufactured using two main methods, solid state processing (e.g., powder metallurgy (PM) and mechanical alloying) and liquid state processing (e.g., stir casting). In the PM method, the matrix and the reinforcing material are mixed, compacted, and subsequently sintered to obtain a consistent material [6,7]. Mechanical alloying is the process in which a reinforcement is introduced into the metal matrix by means of high-energy mechanical milling of the powders to later compact and sinter them, this type of compound is known as dispersion-hardened materials [8,9]. Materials reinforced by dispersion of reinforcing particles belong to the group of composite materials that are manufactured mainly using powder metallurgy (PM) techniques. The convenience of manufacturing MMC using PM was compared with other manufacturing methods by Harris [10]. The manufacture of MMC via the liquid route has received much attention due to its low manufacturing cost, however, the main drawback was that the ceramic particles were randomly distributed and agglomerated during dendrite formation during solidification [11,12]. Using the PM method, it is possible to produce unique MMC materials with a very fine reinforcing particle distribution, which would otherwise be impossible to produce by conventional metallurgy in the liquid state. On the other hand, the methods of mechanical alloying (MA) and mechanical milling (MM) were widely used to produce high-performance materials, such as nanocrystalline materials, nanocomposites, intermetallic compounds, and amorphous materials [13,14]. MA facilitates alloy formation from elemental powders or refinement of pre-alloyed powder grains through high kinetic processing. This occurs due to the repeated welding and cold fracturing of the powders when the impact energy is transferred from the grinding media to the powders, until the stable state is obtained [15,16]. The final particle size (or crystallite) depends on the types of materials used (ductile or brittle), the initial particle size, and the processing conditions used during mechanical milling. Severe plastic deformation, repeated flattening, cold welding, and dust particle fracture during MA/MM lead to significant microstructure refinement, which is often accompanied by a transformation of metastable structures. The final powder produced by MA/MM can be further consolidated by standard powder metallurgy techniques used in the manufacture of bulk materials, or it can be deposited on the surfaces of engineered parts using various thermal spray methods. Some important studies have been carried out to determine the effectiveness of using C as a reinforcing material in the aluminum matrix using mechanical milling [17,18,19]. During the sintering treatment, the content of C dispersed in the aluminum matrix is transformed into the phase Al_4_C_3_, that substantially improves the mechanical properties of the compound [20,21]. Recently, special attention has been paid to Al–Al_4_C_3_ composites. The homogeneous distribution of fine particles of Al_4_C_3_ act as obstacles for the dislocation movement, thus improving the yield stress of the Al matrix. Another important application of these materials is in the improvement of the dry sliding behavior due to residual carbon, which acts as a binder, in the matrix [22,23]. The aim of this study was to determine the combined effect of the C and Cu composition content and the processing conditions on the mechanical properties of the Al matrix by means of statistical analysis. Al–C–Cu composites were fabricated by combining the PM method and mechanical milling (MM) process.

## 2. Materials and Methods

The raw materials were Al powders (99.5% purity, mesh −325) and pre-milled graphite with Cu as additive (C–Cu). Table 1 shows the compositions of metal–matrix composites studied for the first experimental design proposed to determine the effect of C and Cu content on the yield stress (σ_y_) of composites. The table contains the Alp sample, and Alp with different C contents (0, 0.375, 0.75, 1.5, and 3 wt. %) and different Cu contents (0, 0.5, 1, and 2 wt. %). The additive (C–Cu) powders were milled for 4 h, and the Al–C–Cu powders during 1 h. Consolidated samples were sintered for 1 h at 550 °C under vacuum at a heating rate of 50 °C/min. Considering results found in the first experimental design, a second experimental design was established to determine the effect of different processing conditions on the mechanical properties of yield stress (σ_y_) and microhardness (HV) of two composites with low and high C content, 0.75 and 3 wt. % of C respectively. For the experimental design, four sintering conditions process were considered: 1.5 and 3 h of sintering time and 500 and 600 °C of sintering temperature and four milling process conditions were considered: two for the additive powder C–Cu (4 and 8 h of milling time) and two for the Al–C–Cu powder composition (1 and 4 h of milling time). All samples (from first and second experimental design) were mechanically processed in a high energy SPEX mill (Metuchen, NJ, USA) argon was used as the milling atmosphere. Devices and milling media used were made of hardened steel. The milling ball to powder weight ratio was 5:1 and the sample weight was 5 g. For the additive (C–Cu), 1 mg of methanol was used as process control agent and no process control agent was required for the Al–C–Cu composite fabrication. Consolidated products were obtained by pressing the milled powders for two minutes at ≈ 950 MPa in uniaxial load. Consolidated samples were sintered under vacuum at a heating rate of 50 °C/min. Table 2 shows the nomenclature and composition used to fabricate the composite. Table 3 shows the second experimental design: temperature-time sintering and milling time conditions for the 75/25 and 3/100 composites. For the mechanical compression tests, the ratios height. Diameter of 1.0 was used according to the ASTM E9 standard. Using the statistics program “Minitab 17” (State College, PE, USA) (DC Montgomery, 2001), the combined effects of different experimental conditions on yield stress and microhardness were analyzed using contour graphs. To measure the relative density value, the Archimedes’ technique was utilized. The crystallite size, lattice parameters and microstrain of the composites were obtained from the positions of the X-ray diffraction peaks calculated by the Rietveld method.

## 3. Results

### 3.1. Statistical Analysis

The graph of Figure 1a shows the effect of the C and Cu content on the yield stress (σ_y_) of synthesized samples. It was observed that the samples that only contain C (0% Cu) had a significant increase in yield stress. With 3 wt. % of C, the increase was around 208% of the Alp sample. On the other hand, for the samples containing only Cu, the increase was around 33% with 1 wt. % of Cu, in respect to the Alp sample. The graph also shows an increase in the elastic limit with the C–Cu content, however, a definite tendency of the elastic limit was not observed with the C–Cu content. In order to determine if there was a combined effect of the Cu–C mixture, the yield stress values were plotted with respect to the Cu/C ratio. The Figure 1b shows the graph Cu/C ratio as a function of the yield stress (σ_y_). The graph shows that each curve with different C content presents a maximum yield stress value when the Cu/C ratio corresponds to 0.333%, followed by a noticeable decrease of the yield stress after this point. The composite sample with a Cu/C ratio of 1/3, produced an increase in yield stress value of ≈ 233%, in respect to Alp sample.

Figure 2a,b show the results of yield stress (σ_y_) and Vickers microhardness (HV) of the 3/100 and 75/25 samples. As observed, the yield stress and microhardness curves showed similar behavior with each experimental condition used. The 3/100 composites showed the best mechanical properties for each type of test, and the milling time for the synthesis of C–Cu powder and the Al–C–Cu composites had the most significant effect on the mechanical properties. For the yield stress test, the best results found in the 3/100 samples were at 4–4, 1.5 h, and 550 °C experimental conditions; and for the 75/25 samples, the best results were at 4–4, 3 h, and 550 °C experimental conditions (see Figure 2a). In addition, an increase in the milling time in the C–Cu additive powder from 4 to 8 h produced, in most of the samples (3/100 and 75/25), a decrease in the yield stress values. For the microhardness test, the best results found in the 3/100 samples were at 8–4, 1.5 h, and 550 °C experimental conditions; and for the 75/25 samples, were at 4–4, 1.5 h, and 550 °C experimental conditions (see Figure 2b). In the case of the 3/100 samples, an increase in the milling time in the C–Cu additive powder from 4 to 8 h produced an improvement in the microhardness results (see Figure 2b), except for the sample processed at 8–4, 3 h, and 600 °C experimental conditions.

The yield stress (σ_y_) and microhardness contour graphs for 75/25 and 3/100 composites are shown in the Figure 3 and Figure 4. For the case of the 75/25 composites (see Figure 3a), the maximum yield stress values that reached ≈ 250 MPa were at a low temperature of sintering (550 °C) and 3 h of time of sintering. The contour graph also showed that the yield stress increased ≈ 6% when the sintering time changed from 1.5 to 3 h (at hold temperature of 550 °C). On the other hand, the maximum microhardness values obtained of ≈ 123 HV was at a low temperature of sintering (550 °C) and 1.5 h of sintering time (see Figure 3b). For the 3/100 composites, the maximum yield stress value reached of ≈ 345 MPa was at the low sintering temperature (550 °C) and 1.5 h of sintering time (see Figure 4a). It is important to notice that the yield stress increased ≈ 10% when the sintering time changed from 1.5 to 3 h (at 600 °C hold). On the other hand, for the microhardness tests, a change in sintering temperature (from 550 to 600 °C, at 3 h hold) produced an increase of ≈ 18% in the microhardness values (see Figure 4b). In addition, changing sintering time from 1.5 to 3 h (at 600 °C) produced an increase of ≈ 23% in microhardness values, with a maximum value obtained of about 170 HV.

The main rate of drop in mechanical properties with the temperature was also analyzed. For the 75/25 sample, the elastic limit dropped faster (≈ 10%) in samples subjected to a temperature change from 550 to 600 °C (at 3 h hold); while at 1.5 h time of sintering, the yield stress only dropped ≈ 4% (see Figure 3a). On the other hand, for 3/100 samples, the yield stress dropped ≈ 13% when the temperature was increased from 550 to 600 °C (at 1.5 h hold), while in samples subjected at 3 h of sintering time, the change of sintering temperature produced practically no effect on the yield stress (see Figure 4a). On the other hand, microhardness dropped ≈ 15% when 3/100 samples were subjected to a sintering temperature change from 550 to 600 °C (at 1.5 h hold) and dropped ≈ 10% when samples were subjected a change in the time of sintering from 1.5 to 3 h (for samples sintered at 550 °C), as observed in Figure 4b. The 75/25 sample did not show a noticeable change in microhardness values with the change in temperature/sintering time process (see Figure 3b).

The results of the RD measurements for different compositions, milling times, sintering times, and temperature of sintering are indicated in the graph of Figure 5.

In general, the RD curves show similar behavior to that observed in the yield stress and microhardness results, but in the opposite direction. For both samples, 3/100 and 75/25, the combination 4–4 and 8–4 (at different temperature/time of sintering) processing conditions produced the lowest RD values. According to the results, powders with greater mechanical resistance show greater difficulty to be compacted in green, resulting in lower RD values. Samples with high C–Cu content and with high milling time of processing showed greater mechanical resistance (as observed above) and reduced particle size. In addition to this, with a smaller particle size, the pores are smaller, and more pressure is required to collapse them [24]. Contour graphs were obtained using the “Minitab” program to know the effect of sintering temperature (550 and 600 °C) and sintering time (1.5 and 3 h) on the relative density of the composites, using the 4–4 processing condition. The results are shown in Figure 6a,b for the composites 75/25 and 3/100, respectively. The highest rate of sintering was observed when the sintering time changed from 1.5 to 3 h (at 600 °C hold) for both samples, 75/25 and 3/100. An increase in RD of ≈ 1% was observed in 75/25 samples when the sintering time changed from 1.5 to 3 h (at 600 °C hold) as is appreciable in Figure 6a, and about 2.2% for the 3/100 sample under the same conditions (see Figure 6b).

### 3.2. Microstructural Analysis

Figure 7a,b show the metallographic microstructure of 3/100 and 75/25 sample composites, the images show the microstructure and the particle size in the milled and compacted condition. As observed, the 3/100 sample showed a smaller particle size.

The graphs of Figure 8a,b show the cumulative frequency distribution curves of the particle size for Alp and for compounds with 0.75 and 3% C, respectively, as a function of Cu content. The graphs indicate the position of the median, which corresponds to 50% of the cumulative percentage, while the standard deviation corresponds to 84 and 16% (± 1σ) of the cumulative percentage. For the samples with 0.75% C, the particle size decreases with an increase of Cu content from 0.25 to 0.5 wt. %. For the 75/25 sample, 84% of the particles were below 150 µm in size, while 80% of the particles were below 100 µm in size for the 75/50 sample (see Figure 8a). On the other hand, for the composites with high C content (3 wt. %), a clear decrease in grain size was observed with the C content for the 3/0 sample, and ≈ 84% of the grains were smaller than 50 µm in size (see Figure 8b). Similar results were found for the 3/100 sample (≈ 84% of grains were <50 µm), but in this case, unlike sample 3/0, it was observed that the maximum particle size found was around 125 µm, and 200 µm for the 3/0 sample. The 3/200 sample showed an evident increase of particle size, where approximately 84% of the particles were below 70 µm and the maximum particle size found was around 250 µm.

### 3.3. X-ray Diffraction Analysis

Figure 9a,b show the results of the X-ray diffraction analyzes of the samples with low and high C content, in the non-sintered state and the sintered samples respectively.

Figure 10a shows the enlargement of X-ray diffraction profiles. In the patterns, it is appreciable the presence of the Al_4_C_3_ phase for the composites with 75/0, 75/25, and 75/50 samples in the sintered condition. The best refinements of Rietveld fits obtained from the XRD standards, of the samples analyzed before the sintering process, were obtained when the crystallite size and the microstrain were considered as the main broadening effect of the peak. For the samples analyzed after the sintering process, it was when the microstrain was considered the main broadening effect of the peak. An example of Rietveld refinement of an XRD pattern for the sintered sample 75/0 was shown in Figure 10b.

Figure 11a,b show the crystallite size and microstrain from Rietveld refinement analysis. Figure 10a shows that crystallite size results as function of C, for low C and high C content, in the non-sintered condition. As is observed in the graph, the Alp sample showed a reduced crystallite size (≈ 70 nm) as a product of the mechanical milling process. The addition of 0.75 wt. % of C content produced a greater decrease in the crystallite size of about 10%. However, the addition of 0.25 wt. % of Cu content (75/25 sample) resulted in a significant reduction in crystallite size of about 22% with respect to the Alp sample, and a greater decrease of about 33% with 0.5% Cu was observed. These correlated well with grain size results found in metallography images, where the grain size also decreased with the Cu content (see Figure 8a). On the other hand, the addition of 3 wt. % of C content to the Al matrix (see Figure 11a, high C content) produced a significant decrease in crystallite size of about ≈ 65% (with respect to the Alp), and the addition of 1 wt. % of Cu content (3/100 sample) produced a ≈ 55% decrease in crystallite size. However, for the sample with 2% Cu content (3/200 sample), the crystallite increased markedly, even above the value of pure aluminum (≈ 85 nm). Similar behavior was observed in grain size results found in metallography images (see Figure 8b), where the increase in the Cu content at 2 wt. % in Cu content resulted in an increase in grain size. Figure 10b shows the microstrains values results in the sintered and non-sintered condition, as a function of C for low C and high C content. As observed in the graph, the Alp sample shows a relatively high microstrain value of ≈ 0.002, a product of the mechanical milling process (see Figure 11a for low C content). The addition of 0.75 wt. % of C content produced an increase in the microstrain value of about 25%, in respect to Alp sample; while adding 0.25 wt. % of Cu content (75/25 sample) resulted in an increase of about 35%, with respect to Alp sample. With a 0.5% Cu (75/50 sample), a slight decrease in value was obtained of about 30%. After the sintering process, a similar decrease of microstrains values was observed for each sample, except for the sample 75/50 which shows a marked drop with the sintering process (from 0.0026 to 0.0013). For samples with high C (see Figure 11b) content, all the samples show high microstrain values, about 0.0029 for 3/0 sample before sintering and about 0.0025 after sintering. The microstrains values for 3/100 and 3/200 were similar before and after sintering (0.0021 and 0.0025, respectively).

## 4. Discussion

From the second experimental designs results, we observe that increasing the milling time in the fabrication of the Al–C–Cu compound powder from 1 to 4 h significantly increased both the yield stress and the microhardness of the composites. That is, for the 3/100 samples fabricated at 1–4, 1.5 h, and 550 °C experimental conditions, the yield stress was about ≈ 260 Mpa; while the samples fabricated at 4–4, 1.5, and 550 °C experimental conditions, a yield stress value of ≈ 360 Mpa was reached (an increase of ≈ 138%, in respect to the 1–4 experimental conditions). A similar behavior was observed in the microhardness test. The increase of the mechanical properties of the composite with the milling time is related to several strengthening mechanisms: grain size refinement, powder surface area increases, crystallite size reduction, integration and dispersion of the reinforcing particle into the Al matrix, and strain hardening, among others [25,26]. From the results, we observed that an increase in sintering temperature and time also favored the precipitation and growth of second phases (e.g., Al_4_C_3_ or AlCu,); this effect was observed in the contour plot for the yield stress and microhardness tests when the sintering time was increased from 1.5 to 3 h. The presence of these second phases has been previously reported in other works on X-ray diffraction analysis carried out on samples with C and Cu content [27,28]. The most significant effect was observed in the microhardness tests for 3/100 samples, where a change in sintering change from 1.5 to 3 h (at 600 °C hold) resulted in an increase of approximately ≈ 23% in the microhardness value; while, for the yield stress tests (under the same conditions), it presented an increase of ≈ 10% (see Figure 4a,b). The difference between the yield stress and microhardness values obtained can be expected, because in the compression test, the resistance was obtained from the overall particle size of the compact and depends largely on the bond between the particles. In contrast, in microhardness tests, resistance was evaluated for a single particle of the composite powder. The degree of cohesion between adjacent particles strengthens with increasing temperature or sintering time, however, the increase in the yield stress was not proportional to that observed in the microhardness test. From this, it follows that what may be affecting the yield stress results is due to a poor sintering process. The difference between both tests was also observed in the 75/25 samples, where the elastic limit increased ≈ 7.5% when the sintering time changed from 1.5 to 3 h (at 550 °C hold), while the microhardness values (subjected to under the same conditions) suffered a decrease of ≈ 9.5% (see Figure 3a,b). On the other hand, an increase in sintering time and temperature also favors recovery, recrystallization, and grain growth, which negatively affects the mechanical properties of the composite. For the 3/100 samples, the yield stress dropped ≈ 13% in samples subjected to a change of sintering temperature from 550 to 600 °C (at 1.5 h hold), and microhardness dropped ≈ 15% subjected to the same conditions. In this sense, for samples sintered at high temperatures (at 600 °C), the mechanism of recovery, recrystallization, and grain growth are mainly present; while for samples subject to long sintering time (3 h), the mechanism of precipitation of the second phase was mainly present, as stated above. The relative density results confirm the assumptions found above. An increase of ≈ 1 and 2.2% in RD was found in 75/25 and 3/100 samples, respectively, when the sintering time changed from 1.5 to 3 h (at 600 °C hold), so that at high sintering time (3 h), the sintering process mechanism was strengthened. This result means that the yield stress results were affected by low cohesion between grains due to an inefficient sintering process, as observed above. This result may be related to the oxide layer on the Al particles, which prevents atomic mass transport during sintering. This undesirable effect during sintering was previously reported by Randall et al. [29], where the possible formation of oxide during sintering was established, which is greater, the greater the surface area of the powders. The presence of graphite distributed on the surface of compound powder particles during the milling process affecting the sintering process was reported previously [29,30]. It is important to note the phenomena presented above, as mechanism of recovery, recrystallization, grain growth, precipitation, etc, were faster in samples with high C content (3/100 samples), caused by a microstructure with a fine grain size and with high microstrain values, resulting in an accelerated diffusion mechanism. Other authors have reported that smaller particles tend to sinter at a higher rate due to the increased surface area [31]. From the microstructural results, it was observed that the Cu content had an important effect on the grain size. For a low content of C, the grain size decreased as the content of Cu increased (see Figure 8a). However, for high C content composites, the grain size decreases only for a Cu content of 1 wt. %, but for 2 wt. %, the grain size increases markedly. The effect of the Cu content on the crystallite follows the same behavior (see Figure 11a) as that observed in the grain size analyzes shown in Figure 8a (for non-sintered samples). In general, at low C content, an increase in Cu content results in a decrease in grain size and crystallite size. On the other hand, for samples with high C content, an increase in Cu content (>1 wt. %) produces an increase in grain size and crystallite size. This behavior together with the microstrain results may explain the combined effect observed on the yield stress, where at Cu/C ratio of ≈ 0.33, a characteristic peak is observed. The analysis is as follows. The 75/25 sample shows reduced crystallite size and relatively high microstrain values after the milling process and also high microstrain values after the sintering process (compared to the 75/50 sample). On the other hand, the 3/100 sample shows a reduced crystallite size (if compared with the 3/100 sample) and relatively high microstrain values after the milling process and high microstrain values after the sintering process. This combination of reduced crystallite size and high microstrain values (after the sintering process) produced a high yield stress values for samples with Cu/C ratio of ≈ 0.33. On the contrary, for the sample with 2 wt. % Cu content (3/200), a relatively high value of crystallite was observed compared to the crystallite size of that of the 3/100 sample after the milling process, which resulted in a low yield strength value. Additionally, it was observed that, the 3/200 sample also underwent a decrease in microstrain value after the sintering process, below the value of the sample of 3/100.

## 5. Conclusions

From the first experimental design, the following conclusions were derived:A characteristic peak was found in the yield stress test when the Cu/C content ratio of ≈ 0.33 was used to fabricate the composites. This increment of the yield stress was related to the microstructure of the composites after the milling and sintering process.From the microstructural analyses, 3/100 samples mainly showed a smaller crystallite size and a smaller and more homogeneous grain size. The 75/25 samples mainly showed a reduced crystallite size and relatively high microstrain values after the sintering process.

From a second experimental design, the following conclusions were derived:From the contour plot, a significant difference between the microhardness and yield stress was found. From the contour plot analyses carried out on microhardness and yield stress, it was deduced that in the samples sintered at the higher temperature (at 600 °C), the mechanism of recovery, recrystallization, and grain growth were mainly present, which resulted in a decrease in mechanical properties. On the other hand, in the samples subject to long sintering time (3 h), the mechanism of precipitation of second phase was mainly present, resulting in an improvement in mechanical properties.From the difference found between the yield stress and microhardness, it was deduced that yield stress results were affected by low cohesion between grains due to a poor sintering process. These results were corroborated by the RD analyses, and were the best sintering processes that occurred at high sintering time (3 h) experimental condition. However, a small increase in RD was observed.

## Figures and Tables

**Figure 1 materials-14-01969-f001:**
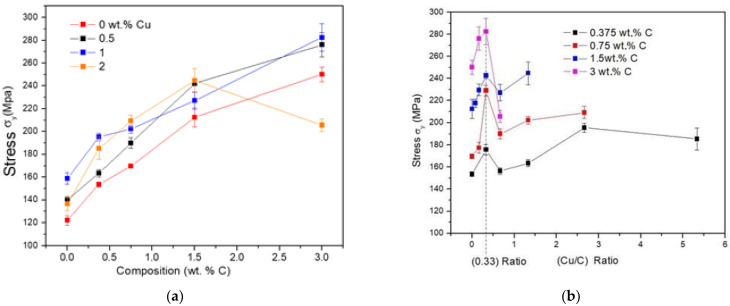
Yield stress (σ_y_) as a function of composition: (**a**) as a function of C and at different Cu content; (**b**) as a function of the Cu/C ratio and at different C content.

**Figure 2 materials-14-01969-f002:**
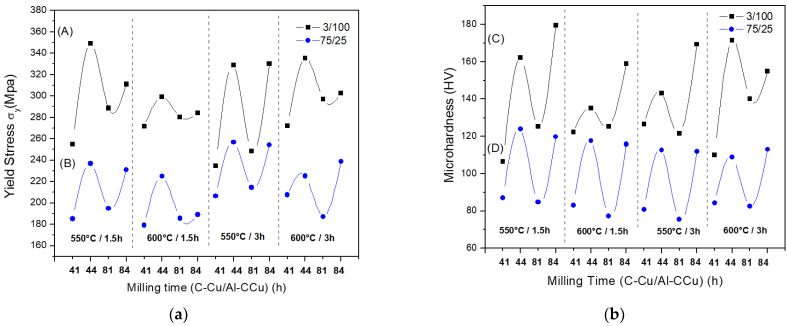
(**a**) Yield stress, σ_y_, as a function of the composition and different processing conditions. (A) 3/100 samples. (B) 75/25 samples. (**b**) Microhardness (Hv) as a function of the composition and different processing conditions. (C) (3/100) samples. (D) 75/25 samples.

**Figure 3 materials-14-01969-f003:**
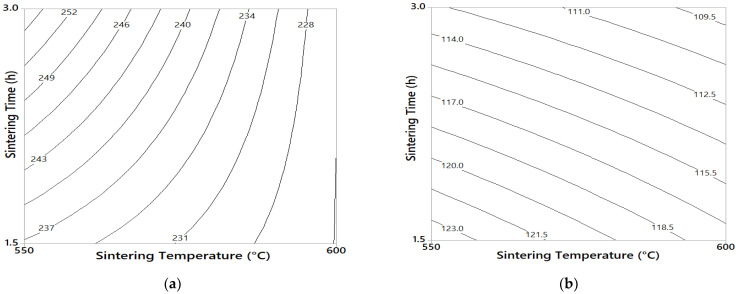
(**a**) Combined effect of time/temperature of sintering for the 75/25 sample on the yield stress (at 4–4 hold conditions). (**b**) Combined effect of time/temperature of sintering for the 75/25 sample on the microhardness (at 4–4 hold conditions.

**Figure 4 materials-14-01969-f004:**
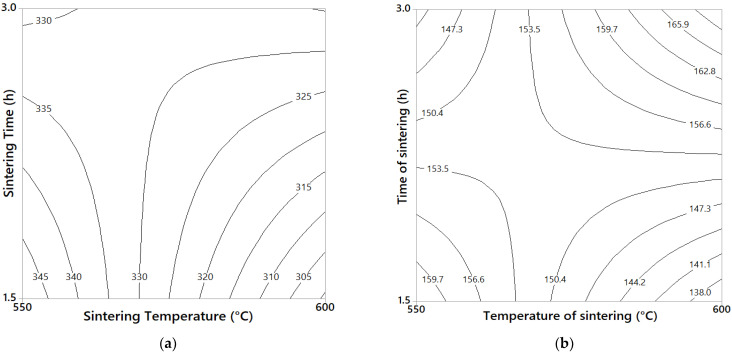
(**a**) Combined effect of time/temperature of sintering for the 3/100 samples on the yield stress (at 4–4 hold conditions). (**b**) Combined effect of time/temperature of sintering for the 3/100 sample on the microhardness (at 4–4 hold conditions).

**Figure 5 materials-14-01969-f005:**
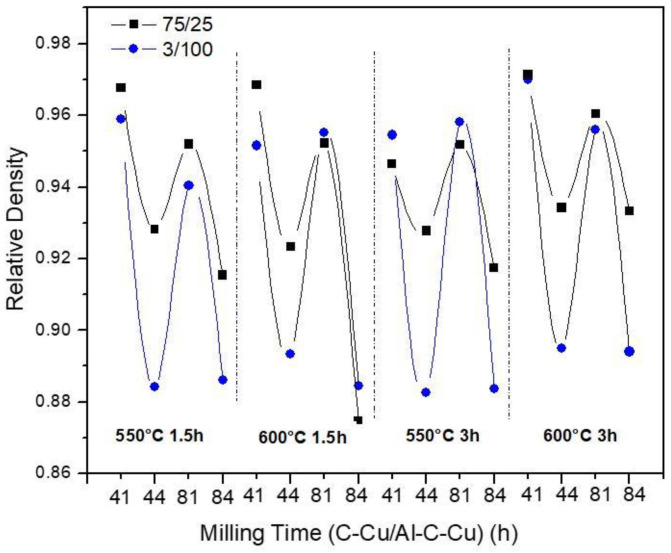
Relative density as a function of composition and processing conditions.

**Figure 6 materials-14-01969-f006:**
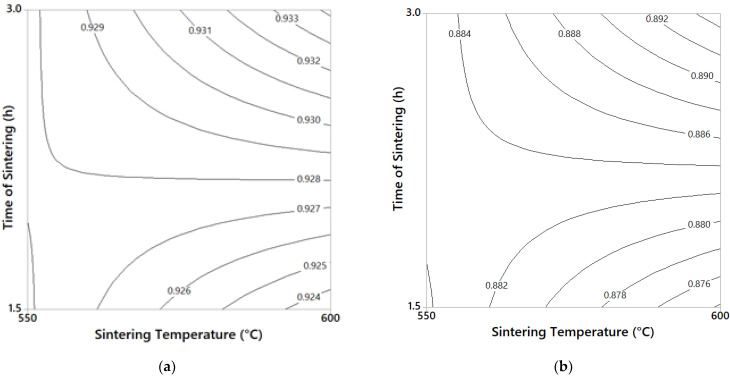
(**a**) Combined effect of time/temperature of sintering for the 75/25 samples on relative density (at 4–4 hold conditions). (**b**) Combined effect of time/temperature of sintering for the 3/100 samples on relative density (at 4–4 hold conditions).

**Figure 7 materials-14-01969-f007:**
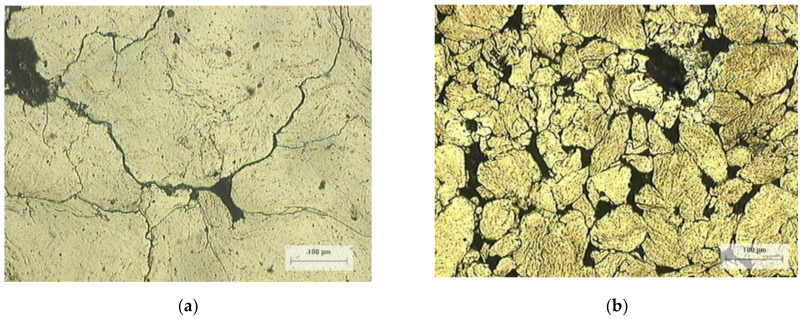
(**a**) Metallographic microstructure of the 75/25 composite. (**b**) Metallographic microstructure of the 3/100 composite.

**Figure 8 materials-14-01969-f008:**
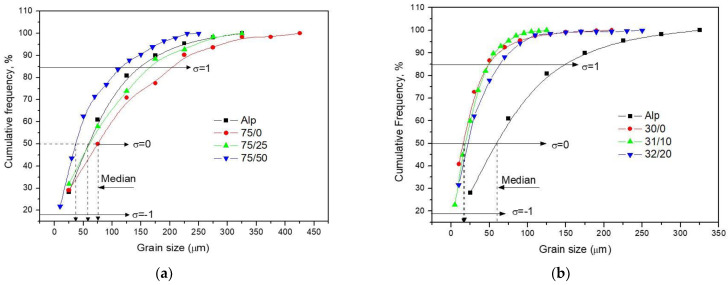
(**a**) Cumulative frequency curves of grain sizes for 75/25 composites. (**b**) Cumulative frequency curves of grain sizes for 3/100 composites.

**Figure 9 materials-14-01969-f009:**
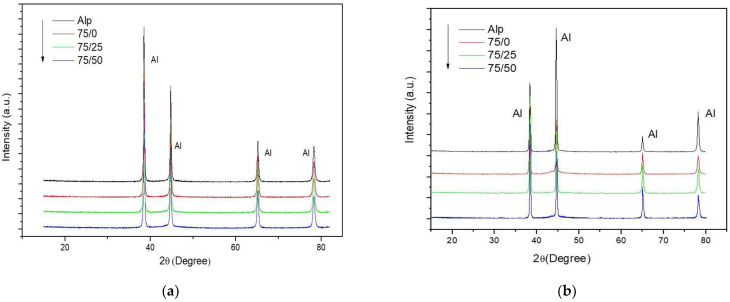
(**a**) X- ray diffraction profiles for the non-sintered composites with low C content. (**b**) X- ray diffraction profiles for the sintered composites with low C content.

**Figure 10 materials-14-01969-f010:**
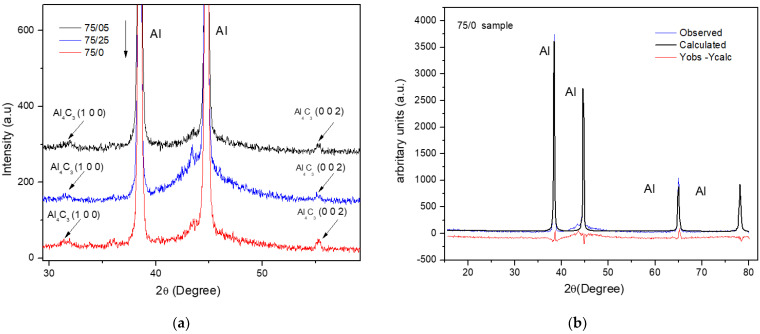
(**a**) Image enlargement of X-ray diffraction profiles of sintered samples. (**b**) The Rietveld refinement of the XRD pattern of the 75/0 sintered sample.

**Figure 11 materials-14-01969-f011:**
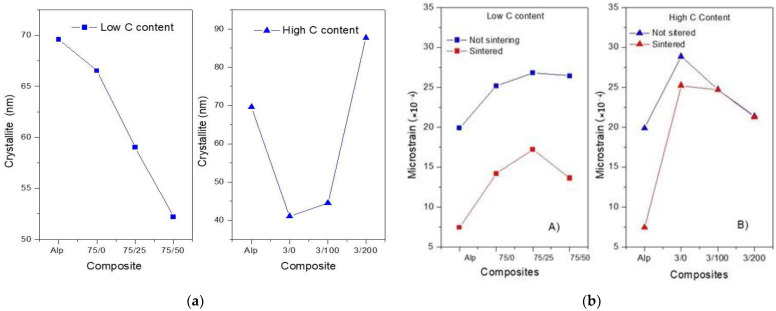
(**a**) Crystallite size for not sintered composites for high C content and low C content. (**b**) Microstrain values for not sintered and sintered composites, for high C and low C content.

**Table 1 materials-14-01969-t001:** Compositions for Al–C–Cu composites fabrication.

C (wt. %)	Cu (wt. %)			
0	0	0.5	1	2
0.375	0	0.5	1	2
0.75	0	0.5	1	2
1.5	0	0.5	1	2
3	0	0.5	1	2

**Table 2 materials-14-01969-t002:** Composition and identification of samples.

	Composition (wt. %)		
Nomenclature	Al	C	Cu
Alp	100	0	0
75/0	99.25	0.75	0
75/25	99.0	0.75	0.25
75/50	98.75	0.75	0.5
30/0	97	3	0
30/100	96	3	1
30/200	95	3	2

**Table 3 materials-14-01969-t003:** Temperature–time sintering and milling time conditions for the 75/25 and 3/100 composites.

Milling Time (h) C–Cu	Milling Time (h) Al–C–Cu	Sintering Temperature (°C)	Sintering Time (h)
4	1	550	1.5
8	1		
4	4		
8	4		
4	1	600	1.5
8	1		
4	4		
8	4		
4	1	550	3
8	1		
4	4		
8	4		
4	1	600	3
8	1		
4	4		
8	4		

## Data Availability

Data sharing is not applicable for this article.

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
