# Peer review of "Statistical and Microstructural Analyses of Al–C–Cu Composites Synthesized Using the State Solid Route"

_materials, 2021, doi:10.3390/ma14081969_

Round 1

Reviewer 1 Report

In the present work, the influence of sintering temperature, sintering time and milling time on the yield stress and hardness of prepared samples has been studied. In Al metal matrix composites, the wettability of C with Al or with Cu is poor, which induces weak bonding interface in the composites. In this work, why did the prepared sample have improved mechanical properties? The authors didn't give clear explanations. In addition, this work didn't provide interesting results. The manuscript is not suitable for publication in Materials.

Author Response

Dear rewiever

We do appreciate your feedback and will use it to evaluate changes and make improvements in this work

Thanks a lot

Reviewer 2 Report

I think the authors should revise the paper according to the following remarks:

  1. The aim of the work should be clearly stated in the Introduction. Also it would be useful for a reader to see possible applications of the materials under investigation shown in the Introduction.
  2. The conclusions also should be rewritten to present the main items (1, 2, …)
  3. X-ray analysis of the specimens with low content of C (Fig 9) does not reveal the presence of Al4C3 and AlCu, which are mentioned for the first time on line 260. Why not to present more X-ray patterns?
  4. In the Introduction the authors refer to two papers [1,2] dealing with Al and Mg matrix composites to support their statement on “high thermal resistance”. What that?
  5. Wording “experimental designs” does not look as the best choice to describe the specimen compositions.
  6. “Nomenclature” 75/25 occurs for the first time in the title of Table 2. Then it is described in Table 3. Should be explained in the text before Table 2.
  7. The Fig 8 caption should be improved: distribution of what?
  8. “increasing the milling time … increased the resistance of the composites” (lines 251 - 252). Should be shown specific properties.
  9. “Rietveld results” on line 221 maybe an expression used in the x-ray community. But for readers outside that community the wording sounds a bit strange.

Remarks 1 to 3 are the main ones.

Reviewer 3 Report

The authors present interesting results which deserve publication in Materials. However, there are some points which can be improved. Here are my main conclusions:

  1. The microstructural analysis is briefly described. Did the authors consider SEM analysis? It might be useful for microstructural analysis. What about EDS/EDX analysis?
  2. XRD data are presented and the Rietveld analysis was done. However, I do not see any refinement results. There are only figures presenting crystallites size and microstrain for various composites. Rietveld refinement is a complicated procedure and refinement factors have to be presented. Also, the detected crystalline phases have to be described.

Round 2

Reviewer 1 Report

The authors didn't answer the questions. I don't recommend publishing the manuscript in Materials.

Author Response

Dear reviewer, I express my sincere apologies.

On line 49, a paragraph was included where it is argued that the use of the mechanical milling technique is used to address the wettability problem of this type of composite.

"On the other hand, previously published works have shown that composite materials Al-C manufactured by the liquid route have poor wettability between molten metal and reinforcing particles. This problem can be overcome using solid-state processing to make the MMNC. The PM route allows several advantages over established casting processes, particularly for incorporating these carbon allotropes into metallic matrices through the ball milling process, considerably improving the metal matrix's mechanical properties [13-15]"

[13] Mary A. Awotundea,Adewale O. Adegbenjo, Babatunde A. Obadele, Moses Okoro, Brendon M. Shongwe, Peter A. Olubambi. Influence of sintering methods on the mechanical properties of aluminium nanocomposites reinforced with carbonaceous compounds: A review. j m a t e r r e s t e c h n o l. 2 0 1 9; 8(2): 2432–2449.

[14] Riccardo Casati and Maurizio Vedani. Metal Matrix Composites Reinforced by Nano-Particles—A Review. Metals 2014, 4, 65-83.

[15] Abdollah Saboori, Seyed Kiomars Moheimani, Mehran Dadkhah, Matteo Pavese, Claudio Badini and Paolo Fino. An Overview of Key Challenges in the Fabrication of Metal Matrix Nanocomposites Reinforced by Graphene Nanoplatelets. Metals 2018, 8, 172.

Reviewer 3 Report

The authors added comments for my previous remarks. However, I still have some further questions.

  1. In the XRD pattern there is an evident peak above 40° which is not indexed. Which crystal phase correspods to such peak ? Did it was considered in Rietveld refinement ?
  2. What about Rietveld refinement factors ? For all refined patters the authors have to add some refinement data, at least refinement factors. It is an indication if refinement was performed correctly or not. Please add such data.
